# Chemical Composition of White Wines Produced from Different Grape Varieties and Wine Regions in Slovakia

**Silvia Jakabová** [1,*], **Martina Fikselová** [1], **Andrea Mendelová** [1], **Michal Ševčík** [2], **Imrich Jakab** [2], **Zuzana Aláčová** [1], **Jana Kolačkovská** [1] and **Violeta Ivanova-Petropulos** [3]

[1]  Faculty of Biotechnology and Food Sciences, Institute of Food Sciences, Slovak University of Agriculture in Nitra, Trieda Andreja Hlinku 2, 94976 Nitra, Slovakia; martina.fikselova@uniag.sk (M.F.); andrea.mendelova@uniag.sk (A.M.); zuzana.alacova47@gmail.com (Z.A.); jkola@centrum.sk (J.K.)

[2]  Department of Ecology and Environmental Sciences, Faculty of Natural Sciences, Constantine the Philosopher University in Nitra, Trieda Andreja Hlinku 1, 94974 Nitra, Slovakia; msevcik@ukf.sk (M.Š.); ijakab@ukf.sk (I.J.)

[3]  Faculty of Agriculture, University "Goce Delčev"—Štip, Krste Misirkov 10-A, 2000 Štip, North Macedonia; violeta.ivanova@ugd.edu.mk

*   Correspondence: jakabova@is.uniag.sk

**Abstract:** In this work, chemical parameters such as sugar (glucose and fructose) content, organic acid (total acids, malic and tartaric acids), total phenolic content and the antioxidant activity of 12 white wines (chardonnay, pinot blanc and pinot gris) from various wine regions in Slovakia were studied in order to identify differences among the varieties and wine-growing regions. The wine samples were examined by Fourier-transform infrared spectroscopy (FTIR) and UV-VIS spectrophotometry (for determination of total polyphenolic content (TPC) and total antioxidant activity (TAA)) methods. Content of alcohol ranged between 11.50% and 13.80% with the mean value 12.52%. Mean content of total acids varied between $4.63 \pm 0.09$ and $6.63 \pm 0.05$ g.L$^{-1}$, tartaric acid varied between $1.62 \pm 0.09$ and $2.93 \pm 0.03$ g L$^{-1}$, malic acid was found in the concentrations ranged from $0.07 \pm 0.05$ and $2.50 \pm 0.08$ g L$^{-1}$ and lactic acid was present between 1.53 and 0.01 g L$^{-1}$. The content of fructose was, in general, higher in the samples from the Južnoslovenská and Nitrianska wine regions and glucose was higher in the Malokarpatská wine region. Chardonnay wines showed the highest content of total polyphenols and the antioxidant activity in the samples ranged from $51.06 \pm 027$ to $72.53 \pm 0.35\%$ inhibition of DPPH. The PCA analysis based on chemical descriptors distinguished the Nitrianska and Stredoslovenská wine regions. According to similarities among the wine samples, four main classes were formed by cluster analysis.

**Keywords:** white wine; FTIR; UV-VIS spectrophotometry; chemical composition; geographic indication

## 1. Introduction

The primary and secondary plant metabolites play critical roles in the health of humans and could be nutritionally important. Many of these compounds are also considered to be crucial factors for food and beverage quality. Evaluation of food and wine quality and the value for human health is often connected with various physical-chemical and organoleptic properties [1–3]. Wine, as one of the world's widely consumed beverages, has been proved to have health-promoting effects as a function of metabolites belonging to various different groups of chemicals [1,4].

Various factors affect the quality of wine, including its taste and aroma, such as grape variety, ripening stage, environmental conditions, soil, vine cultivation, as well as winemaking practices applied during wine production [5–10]. The main difference between red and white wine production is the duration of maceration applied. For red wine production, studies are primarily focused on the influence of different maceration times/days on the extraction of grape pigments and tannins [11]. For white wine production, maceration

is kept to a minimum and seldom lasts more than a few hours in order to avoid extensive contact with oxygen, which can cause browning of the wine and deterioration of the overall quality [12]. Usually, the grape juice runs freely from the crushed grapes followed by the immediate addition of $SO_2$ in order to protect enzymatic oxidation. Moreover, wine possesses antioxidant properties which are attributed by the content of biologically active compounds, such as phenolic compounds and organic acids [13,14]. High antioxidant activity of the polyphenolic compounds is linked with both free radical scavenging and transition metal chelating properties that contribute to antibacterial, antimutagenic, anti-inflammatory and vasodilatory action [15,16].

Determination of the main chemical composition of wine, including ethanol content, residual sugars, total and volatile acidity, main organic acids, as well as aroma compounds, phenolic compounds and antioxidant properties could be considered as one of the most important parameters that determine the quality of wine. Carbohydrates are primary metabolites that are used by plants in the process of respiration and as a building material for cell wall structure. They are synthesized throughout the process of photosynthesis and their content increases as fruits ripen. Main carbohydrates in grapes are glucose and fructose, also known as reductive sugars, with concentrations of 150–250 g $L^{-1}$ in must. Impact of these components in wines, increase its viscosity, is important qualitative attributes of wine [17]. In grapes, glucose and fructose are present in almost equal concentrations. The glucose/fructose ratio is mainly affected by the climate conditions of the region and the year of harvest and, usually, the ratio decreases during warmer weather and increases in colder weather conditions [6–8]. During fermentation, the glucose/fructose ratio decreases from 0.95 at the beginning to 0.25 towards the end of fermentation since glucose is primarily fermented because it is used by the yeast. The composition and content of sugars have been proposed by Gnilomedova et al. [6] as one of the chemical characteristics that can be used for verifying the authenticity of the grape origin of various products. In order to check the origin of wine and its adulteration, such as unauthorized addition of sucrose, the nuclear magnetic resonance is used [18].

Over the last few years, higher attention has been paid to organic acids, their importance and their health benefits. Organic acids are natural components that contribute to the organoleptic (flavour, colour and aroma) and healthy properties (antioxidant and antimicrobial activity) of food [14,19–21]. In wines, organic acids have a major role in the composition, the stability and the organoleptic qualities [14,22], as well as contributing to preservative properties' enhancement in terms of the microbiological and the physicochemical stability [23]. The main organic acids in wine are tartaric, malic, citric, lactic, succinic and acetic acids. Over 90% of organic acids in grapes and thus in wine are dedicated to malic and tartaric acids. As these two acids are the most abundant in grapes, their levels are often used to determine the date of harvest, since each acid behaves differently during the ripening process [24]. Tartaric acid is the main organic acid in grapes and wines that significantly affects the total acidity of wines. Tartaric acid is the dominant organic acid in wines, which plays a significant role in maintaining the chemical stability of the wine, its colour and its taste. The content of tartaric acid decreases during the fermentation as a result of precipitation in the form of tartaric crystals [25].

The other important chemical parameters which determine wine quality are phenolic compounds. They strongly contribute to the colour, mouthfeel, astringency and bitterness of the wine [16,26,27]. Phenolic compounds originate from different parts of the grape: (i) grape skins contain anthocyanins, flavan-3-ols, flavonols, dihydroflavonols, hydroxycinnamoyl tartaric acids, hydroxybenzoic acids and hydroxystilbenes; (ii) flavan-3-ols and gallic acid are dominant in the seeds; and (iii) hydroxycinnamoyl tartaric acids are mainly present in the juice [28]. The phenolic composition of wines depends on the grape composition, on their extraction into the grape juice, and also, on the subsequent reactions occurring during the vinification, post-fermentation treatments, and wine aging [12]. The content of phenolic substances in the peel can be up to 2.5%, while their proportion is higher in red varieties of grape compared to white varieties [29].

During the last decade many studies have been performed in the analysis of phenolic compounds in wines, as well as organic acids and sugars. With regard to Slovak white wines, limited data on the total phenolics and antioxidant properties are available [30–32]. By current knowledge, there is no similar survey of white wines in Slovakia that reports such a comparison of white wine varieties according to potential chemical descriptors in relation to geographic origin. Therefore, the objectives of the present work were (1) to determine the main carbohydrates (glucose and fructose), organic acids (malic, tartaric and lactic) as well as alcohol content, total phenolics and antioxidant activity in white Slovak wines from chardonnay, pinot blanc and pinot gris varieties, and (2) to study the relationship between the chemical composition of white wines from various varieties and wine geographic indication, applying Fourier-transform infrared (FTIR) spectroscopy and UV-Vis spectrophotometric (TAA, TPC) methods.

## 2. Materials and Methods

### 2.1. Chemicals and Reagents

Folin–Ciocalteu phenol reagent (Centralchem, Slovakia), sodium carbonate p.a. (99%; Centralchem, Slovakia), gallic acid (3, 4, 5-trihydroxybenzoic acid monohydrate, 99%; Alfa Aesar Thermo Fisher (Kandel) GmbH, Kandel, Germany) were used for determination of the total polyphenolic content. All the reagents were dissolved in distilled water. The crucial reagent used for the total antioxidant activity (TAA) measurements was 1,1-diphenyl-2-picrylhydrazyl radical (DPPH) (Sigma-Aldrich; Merck KGaA, Darmstadt, Germany) dissolved in methanol p.a. (99.8%; Centralchem, Slovakia). Standard wine for FT-IR analysis (Bruker Optic GmbH, Ettlingen, Germany) was purchased from OK Servis BioPro, Ltd. (Praha, Czech Republic) and deionized water was used for the sample and mobile phase preparation.

### 2.2. Wine Samples

Samples of 12 different white wines, belonging to three grape varieties of chardonnay, pinot blanc and pinot gris, were analysed in this study. Wines were produced from the 2018 vintage year; all samples were obtained in June 2020 from wineries as three bottles of the same batch of each wine variety. All wines (750 mL glass bottles) were collected from four wine-growing regions in Slovakia, as is shown in Table 1. The wine samples in this study are described in Table 1. Samples were kept at 4 °C before the analysis. All analyses were performed in July 2020 in one day. Location of wine manufacturers is shown in Figure 1.

### 2.3. Sample Preparation

Before analysis, minimal sample preparation was performed, including centrifugation of wine samples for 2 min with a relative centrifugal force 3622. The centrifugate was transferred into another flask and analysed.

### 2.4. FTIR Analyses

Determination of the selected chemical parameters (alcohol content, total acids, tartaric, malic, lactic acids glucose and fructose) in wine samples was performed using the ALPHA Bruker Optik GMBH analyser. Fourier transform infrared spectroscopy (FT-IR) with the attenuated total reflect (ATR) measurement procedure was used. The method allows simultaneous analysis of different parameters within one measurement. The analyser was calibrated for wine with use of calibration data from wine measurements, what was performed by technical support of OK Servis BioPro, Ltd. Company. The calibration data consisted of over 2000 red and white wines. The calibration data containing measurement of calibration spectra and the calibration settings according to the reference values were provided by the accredited (DAkkS) Institute Heidger (Osann Monzel, Germany). The root mean square error of prediction (RMSEP) has been determined. The ranges of calibration curves and RMSEP (in the brackets) were as follows: alcohol 0.12–20.48% (0.27); fructose 0.1–111.5 g L$^{-1}$ (0.71); glucose 0.2–125.6 g L$^{-1}$ (0.87); total acids 2.9–13.5 g L$^{-1}$ (0.31); malic

acid 0.1–4.7 g $L^{-1}$ (0.40); tartaric acid 0.5–5.4 g $L^{-1}$ (0.43); lactic acid 0.0–4.3 g $L^{-1}$ (0.31), for Fourier-transform infrared (FTIR) spectroscopy. Before starting the measurement, the flow-through cell of the ALPHA analyser was rinsed with deionized water and a blank (deionized water) was measured. Approximately 20 mL of wine sample was injected into the flow-through cell, where the sample was heated to 40 °C and then measured. The spectrum was scanned in the wavenumber range between 4000 and 400 $cm^{-1}$. The instrument evaluated the sample within 70–100 s and the results for all five parameters were recorded. Precision of the method was checked by analysing the standard wine for analysis (Bruker Optic GmbH, Ettlingen, Germany) with reference values of individual parameters that were tested by reference methods (high performance liquid chromatography, pH electrode, titration, density meter).

**Table 1.** Wine samples.

| Variety | Sample Abbreviation | Wine Type | Wine Region | Winery |
|---|---|---|---|---|
| Chardonnay | CH 1 | dry | Južnoslovenská | Wine Manufacturer A |
| Chardonnay | CH 4 | dry | Stredoslovenská | Wine Manufacturer B |
| Chardonnay | CH 7 | dry | Malokarpatská | Wine Manufacturer C |
| Chardonnay | CH 10 | dry | Nitrianska | Wine Manufacturer D |
| Pinot Blanc | PB 2 | dry | Južnoslovenská | Wine Manufacturer A |
| Pinot Blanc | PB 5 | dry | Stredoslovenská | Wine Manufacturer B |
| Pinot Blanc | PB 8 | semi-dry | Malokarpatská | Wine Manufacturer C |
| Pinot Blanc | PB 11 | semi-dry | Nitrianska | Wine Manufacturer D |
| Pinot Gris | PG 3 | dry | Južnoslovenská | Wine Manufacturer A |
| Pinot Gris | PG 6 | dry | Stredoslovenská | Wine Manufacturer B |
| Pinot Gris | PG 9 | semi-dry | Malokarpatská | Wine Manufacturer C |
| Pinot Gris | PG 12 | dry | Nitrianska | Wine Manufacturer D |

Note: CH—chardonnay, PG—pinot gris, PB—pinot blanc, numbers are in relation to wine regions: 1–3 Južnoslovenská, 4–6 Strednoslovenská, 7–9 Malokarpatská, 10–12 Nitrianska wine region.

### 2.5. Spectrophotometric Analyses

A double-beam spectrophotometer (T80 UV/VIS Spectrometer; PG Instruments Ltd., OK Service, The Czech Republic) equipped with cuvette holder for 8 cuvettes was used for determination of total polyphenolic content and total antioxidant activity. Glass cuvettes (type S/G/10; Exacta+Optech GmbH, Munich, Germany) were used for analysis. Injected volume of each wine sample for TPC was 50 μL and for TAA 100 μL.

### 2.5.1. Determination of Total Polyphenolic Content

Total polyphenolic content (TPC) was determined according to the modified Folin–Ciocalteu spectrophotometric method [33]. The method is based on the reaction of polyphenols with the Folin–Ciocalteu phenol reagent what is accompanied by formation of blue product in the presence of basic medium as a result of reduction of molybdenum and tungsten oxidation state of 6+ by the hydroxyl group from phenols in the alkaline conditions leading to blue-coloured product [34]. Quantification of polyphenols is based on measurement of the intensity of blue colour. As a representative of polyphenolic compounds, a gallic acid (GA) is often used to express the total quantity of polyphenols in wine. A stock solution was prepared from 0.1 g of GA and its dilution with demineralized water up to 100 mL volume. Then the stock solution was appropriately diluted with demineralized water to prepare calibration solutions in the range of 5–200 mg $L^{-1}$ of GA. The blank was prepared with use of Folin–Ciocalteu reagent and distilled water instead of standard or sample. The calibration curve obtained had correlation coefficient R = 0.999. The concentration of polyphenols was quantified as mg $L^{-1}$ of GA equivalent (GAE). 50 μL of the wine was introduced into 50 mL volumetric flasks, then 2.5 mL of Folin–Ciocalteu reagent diluted with demineralized water (1:2 *v/v*) was added followed by addition of 5 mL $Na_2CO_3$ (20% water solution). Mixtures were vigorously shaken and filled up with

demineralized water to 50 mL volume. The blue-coloured complex was developed in 2 h at laboratory temperature. Each wine sample was used for preparation of triplicates. Measurement was performed with use of the UV/Vis spectrophotometer at 765 nm.

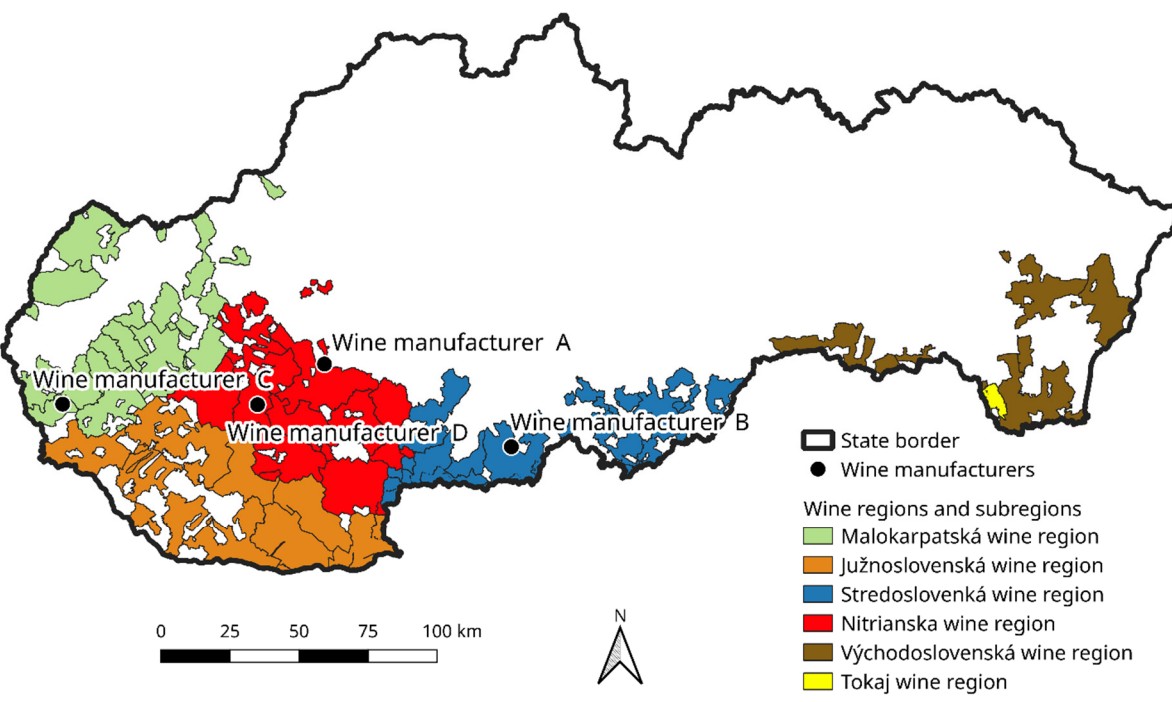

**Figure 1.** Location of wine manufacturers.

### 2.5.2. Determination of Total Antioxidant Activity (TAA)

A Brand-Williams et al. method [35] with a slight modification was used for determination of the total antioxidant activity (TAA). Free radical scavenging assay with use of DPPH by was used for this determination. The principle of this approach lies in the reduction of purple coloured radical 2,2-diphenyl-1-picrylhydrazyl (DPPH) to the yellow coloured derivatives. The DPPH methanol stock solution was prepared from 0.025 g DPPH and its dilution in solvent to provide the final volume 100 mL. Before the measurement, the DPPH solution was diluted with methanol in the ratio 1:9 to provide an absorbance of approx. 0.7. A glass cuvette was filled up with 3.9 mL of the diluted DPPH and initial DPPH absorbance ($A_o$) was read at 515.6 nm. A wine sample was added in amount 100 µL into the cuvette and the content of the cuvette was stirred properly. After 10 min of keeping the cuvettes in the dark, a final absorbance ($A_t$) was measured again. The decrease of absorbance was expressed as the % of inhibition of DPPH radical (Equation (1)). Each wine sample was measured in triplicates and the results were shown as means ± sd.

$$\% \text{ inhibition of DPPH} = \frac{(A_o - A_s) - (A_t - A_s)}{(A_o - A_s)} \cdot 100 \tag{1}$$

where $A_o$ is initial absorbance of DPPH solution, $A_s$ is absorbance of methanol, $A_t$ is absorbance after 10 min. after introducing the wine sample.

### 2.6. Statistical Analysis and Visualisation

Basic descriptive statistics were performed on the data for alcohol content, total acidity, tartaric acid, malic acid, lactic acid, glucose, fructose, TPC and AA, including mean with a standard deviation and Pearson correlation between each chemical parameter. Individual wines were classified by hierarchical clustering using Euclidean distance among the records and Ward's cluster analysis method (Ward v2) of joining the clusters. The dendrogram prepared revealed the distinction of four main groups. To summarize and

visualize the variation between the regions we used principal components analysis (PCA) calculated from centred and standardized data. The ANOVA for repeated measurements was applied for testing differences among the individual chemical parameters in individual wine-growing regions, followed by post hoc comparations using the Tukey HSD method. The results for each wine parameter were standardized and centred based on variety. The differences were tested at the significant level $p < 0.05$. All statistical analyses were performed in the R statistical environment [36].

The Figure 1 was created using QGIS software (3.14.16-Pi, producer: Free Software Foundation, Inc., Boston, MA, USA) [37] to visualize locations of wine manufacturers.

## 3. Results and Discussion

Table 2 shows the chemical composition of all wines according to the different grape varieties (chardonnay, pinot blanc and pinot gris) and the wine regions of origin (Južnoslovenská, Stredoslovenská, Malokarpatská and Nitrianska).

**Table 2.** Characterization of chemical composition of monovarietal wines.

| Wine Sample | Alcohol | Total Acids | Tartaric Acid | Malic Acid | Lactic Acid | Fructose | Glucose | TPC | TAA |
|---|---|---|---|---|---|---|---|---|---|
| | [%] | [g L$^{-1}$] | [g L$^{-1}$] | [g L$^{-1}$] | [g L$^{-1}$] | [g L$^{-1}$] | [g L$^{-1}$] | [mg L$^{-1}$ GAE] | [% Inhibition DPPH] |
| CH1 | 12.23 ± 0.09 | 6.37 ± 0.05 | 2.61 ± 0.02 | 2.50 ± 0.08 | 0.31 ± 0.06 | 6.10 ± 0.08 | 1.33 ± 0.05 | 403.64 ± 9.41 | 59.43 ± 0.30 |
| PB2 | 12.50 ± 0.00 | 6.63 ± 0.05 | 2.78 ± 0.04 | 2.43 ± 0.12 | 0.36 ± 0.06 | 4.8 ± 0.14 | 1.03 ± 0.17 | 283.19 ± 8.03 | 60.74 ± 0.32 |
| PG3 | 12.40 ± 0.00 | 5.43 ± 0.05 | 2.24 ± 0.01 | 1.87 ± 0.05 | 0.52 ± 0.08 | 2.17 ± 0.12 | 2.40 ± 0.16 | 307.77 ± 20.47 | 67.40 ± 0.44 |
| CH4 | 12.20 ± 0.00 | 5.47 ± 0.05 | 2.93 ± 0.03 | 0.07 ± 0.05 | 1.30 ± 0.02 | 0.63 ± 0.12 | 0.30 ± 0.08 | 329.89 ± 9.41 | 60.32 ± 1.03 |
| PB5 | 12.37 ± 0.05 | 5.77 ± 0.05 | 2.56 ± 0.06 | 1.03 ± 0.19 | 0.98 ± 0.08 | 0.53 ± 0.25 | 0.23 ± 0.17 | 379.06 ± 4.92 | 60.96 ± 0.57 |
| PG6 | 11.90 ± 0.00 | 5.77 ± 0.05 | 2.77 ± 0.04 | 0.77 ± 0.12 | 0.82 ± 0.03 | 0.73 ± 0.05 | <0.20 | 268.44 ± 8.03 | 72.53 ± 0.35 |
| CH7 | 12.70 ± 0.00 | 5.23 ± 0.05 | 2.32 ± 0.06 | 1.50 ± 0.08 | 0.86 ± 0.08 | 1.7 ± 0.08 | 0.83 ± 0.05 | 339.72 ± 9.41 | 61.60 ± 0.24 |
| PB8 | 12.20 ± 0.00 | 5.23 ± 0.05 | 2.09 ± 0.07 | 1.10 ± 0.14 | 1.11 ± 0.04 | 4.63 ± 0.09 | 3.33 ± 0.05 | 312.68 ± 8.03 | 59.98 ± 0.51 |
| PG9 | 11.50 ± 0.00 | 5.67 ± 0.05 | 2.57 ± 0.03 | 1.17 ± 0.05 | 0.87 ± 0.05 | 5.33 ± 0.09 | 3.03 ± 0.09 | 256.15 ± 9.41 | 51.06 ± 0.27 |
| CH10 | 13.80 ± 0.00 | 5.10 ± 0.001 | 1.62 ± 0.09 | 1.00 ± 0.08 | 1.53 ± 0.03 | 6.80 ± 0.14 | <0.20 | 396.26 ± 23.41 | 71.87 ± 0.95 |
| PB11 | 12.77 ± 0.05 | 4.93 ± 0.05 | 2.04 ± 0.08 | 1.87 ± 0.05 | 0.14 ± 0.03 | 6.73 ± 0.09 | 0.3 ± 0.16 | 283.19 ± 8.03 | 52.78 ± 0.47 |
| PG12 | 13.67 ± 0.05 | 4.63 ± 0.09 | 1.63 ± 0.06 | 1.63 ± 0.12 | 0.85 ± 0.05 | 1.63 ± 0.09 | 0.67 ± 0.12 | 381.51 ± 8.03 | 66.80 ± 0.66 |
| Mean | 12.52 | 5.52 | 2.35 | 1.41 | 0.8 | 3.48 | 1.13 | 328.46 | 62.12 |
| Min | 11.50 | 4.63 | 1.62 | 0.07 | 0.01 | 0.53 | 0.01 | 256.15 | 51.06 |
| Max | 13.80 | 6.63 | 2.93 | 2.50 | 1.53 | 6.80 | 3.33 | 403.64 | 72.53 |

Abbreviation of wines: CH—charodnnay, PB—pinot blanc and PG—pinot gris. Numbers are in relation to wine regions: 1–3 Južnoslovenská, 4–6 Strednoslovenská, 7–9 Malokarpatská, 10–12 Nitrianska wine region. Results are mean values of three repetitions ± s$_D$ (standard deviation).

### 3.1. Influence of Variety

Total acid content is defined as the concentration of organic acids in the grape or wine [38]. Total acids in white wine samples varied from 4.63 ± 0.09 to 6.63 ± 0.05 g L$^{-1}$.

The content of organic acids, tartaric and malic, was studied in white wines as they are considered as one of the important descriptors of wine. Tartaric acid (2,3-dihydroxybutanedioic acid) is a predominant acid in wines and represents about 50% of total acids in wine [39]. In our wines, the concentration of tartaric acids was similar.

Thus, tartaric acid in chardonnay wines ranged from 1.63 to 2.93 g L$^{-1}$ (mean: 2.37 g L$^{-1}$). Pinot blanc wines contained tartaric acid in a range of 2.04 to 2.78 g L$^{-1}$ (mean: 2.37 g L$^{-1}$) and pinot gris wines presented 1.63–2.77 g L$^{-1}$ (mean: 2.30 g L$^{-1}$). During the fermentation and aging process, its concentration decreases as a result of the formation of tartrates.

The content of malic acid is usually highest at the beginning of the alcoholic fermentation, and afterwards it converts into lactic acid, spontaneously or in the presence of malolactic bacteria, during the malolactic fermentation. During this process, the content of malic acid decreases and the content of lactic acid increases in wine [40]. In our study, most of the wines contained malic acid in concentrations higher than 1 g L$^{-1}$, except wines CH4 and PG6, which contained 0.07 and 0.77 g L$^{-1}$, respectively, which means that malolactic fermentation was almost completed in these wines. In addition, chardonnay wines contained from 0.07 to 2.50 g L$^{-1}$ malic acid (average: 1.27 g L$^{-1}$), in pinot blanc malic acid was present in a range of 1.03–2.43 g L$^{-1}$ (average: 1.61 g L$^{-1}$) and in pinot gris

wines from 0.77–1.87 g L$^{-1}$ (average: 1.36 g L$^{-1}$). Ailer [18] reported that content of malic acid in white and rosé wines is usually in the range between 1–3 g L$^{-1}$. Content of lactic acid was in most cases below 1 g L$^{-1}$, except wines CH4, PB8 and CH10. The obtained results for total acids, tartaric and malic acids in the white Slovak wines were comparable to those reported in the literature for wines from various regions, including Spain, Brazil, Chile, France [41–44].

Glucose and fructose are the main fermentable sugars in wine must. During alcoholic fermentation, yeasts convert most of the glucose and fructose present into alcohol and $CO_2$. Grape musts contain equal amounts of glucose and fructose, and during fermentation glucose is consumed at a higher rate than fructose, which leads to an increased proportion of fructose as fermentation progresses, as was also observed in our wines. In general, all wines contained a higher amount of fructose (on average: 3.81 g L$^{-1}$ for chardonnay wines; 4.17 g L$^{-1}$ for pinot blanc wines and 2.47 g L$^{-1}$ for pinot gris wines) compared to glucose (on average: 0.63 g L$^{-1}$ for chardonnay wines; 1.22 g L$^{-1}$ for pinot blanc wines and 1.53 g L$^{-1}$ for pinot gris wines). The higher values for fructose compared to glucose, obtained for the wine samples, can be explained by the preference of *Saccharomyces cerevisiae* yeasts to ferment glucose, resulting in larger residual amounts of fructose in wines. Restani et al. [45] reported that fructose and glucose levels in high quality wines is low (0.71 ± 0.73 and 0.32 ± 0.44 g L$^{-1}$, respectively), however, in some sweet and sparkling wines fructose and glucose levels can be much higher. Coelho et al. [46] reported the mean glucose and fructose contents were less than 1.23 and 4.97 g L$^{-1}$.

Content of alcohol in wines varied between 11.50 and 13.80% with the mean 12.52%. Mean content of alcohol in chardonnay wines was 12.73 ± 0.65%, in pinot blanc 12.46 ± 0.21% and in pinot gris 12.37 ± 0.82%. The highest content of alcohol was observed in chardonnay from Nitrianska w.r. and the lowest in pinot gris from Malokarpatská w.r.

White wines typically show lower content of phenolic compounds compared to red ones. The most dominant polyphenols in white wines are hydroxycinnamic acid derivatives, hydroxybenzoic acids, flavonols and flavan-3-ols that are related to wine sensorial properties, such as the chromatic characteristics, colour stability, bitterness and astringency [27]. Our wines presented relatively high values of total polyphenols, ranging from 329 to 403 mg GAE L$^{-1}$ for chardonnay, from 283 to 379 mg GAE.L$^{-1}$ for pinot blanc and 256 to 381 mg GAE L$^{-1}$ for pinot gris wines. The highest content of total polyphenols was found in chardonnay from the Južnoslovenská wine-growing region (403.64 ± 9.41 mg GAE L$^{-1}$). In all other chardonnay wine samples, the TPC levels exceeded a content of 300 mg GAE L$^{-1}$. In comparison with the chardonnay wines from Montenegro (TPC 226 mg GAE L$^{-1}$) [47], the content in Slovak chardonnay wines was higher in more than 100 mg GAE L$^{-1}$. Our results are also in agreement with experiments by Bajčan et al. [30] and Čéryová et al. [31], who examined the antioxidant activity, total phenolic and flavonoid contents in monovarietal wines of Welschriesling, chardonnay. According to their results, Slovak white wines were high in polyphenols (average content was 303.2 mg GAE.L$^{-1}$ in Welschriesling, resp. 355.6 mg GAE L$^{-1}$ in chardonnay).

Total antioxidant activity presents one of the important characteristics of wine. In general, a higher level of antioxidants is connected with higher levels of polyphenols. In our study, all wines presented relatively high values for the antioxidant activity, ranged between 51.06 and 72.53% of inhibition of DPPH (average: 63.30% for chardonnay, 58.62% for pinot blanc and 64.45 for pinot gris wines). Our results for antioxidant activity were correlated with previously published data for white wines [30,31,48]. According to several studies, polyphenolic compounds have shown a different behaviour towards DPPH free radical, both in terms of capacity and rate of scavenging [49,50]. The antioxidant activity of phenolic compounds is related with their chemical structure. Higher activity has been reported in the compounds with a high number of hydroxyl groups. The contribution of each polyphenol to the antioxidant activity of wines is different, so the activity of wines depends on their phenolic profile. De Quirós et al. [50] examined correlation between the several polyphenol compounds and their antioxidant activity. A good positive correlation was

observed between antioxidant activity and quercetin content, and rutin and procyanidin B1 showed a reasonable correlation (approx. 0.55). No correlation was found between caftaric acid, the predominant major phenolic compound in wines. In our study, some wine samples showed different trend connected with high TPC values and lower TAA or lower TPC and higher TAA values, what was confirmed in all three batches of wines. A phenolic profile and presence of other kinds of antioxidants in wines was not followed.

To quantify the relationship among each of the chemical parameters, Pearson correlation was used. Weak positive correlation was found between TPC and TAA (r = 0.33), glucose and fructose content (r = 0.25) and malic acid and fructose (r = 0.47). The highest positive correlation was observed between total acids and the content of tartaric acid (r = 0.76) due to the fact that most wine acidity is attributed to tartaric acid concentration. High positive correlation was also observed between alcohol content and TPC (r = 0.63) and alcohol content and TAA (r = 0.44). Weak negative correlation was determined between the TPC and tartaric acid (r = −0.39), similar weak negative correlation was observed between TAA and fructose (r = −0.36), TAA and glucose (r = −0.39), and TPC and glucose (r = −0.33). High negative correlation was observed between alcohol content and total acids (r = −0.54) and the highest was negative correlation between alcohol content and tartaric acid (r = −0.8) and between malic acid and lactic acid (r = −0.81).

### 3.2. Influence of Wine Region

In terms of the geographical origin of wine, certain regions are particularly famous for their wines. Elemental fingerprinting presents one attempt to measure regionality [51]. The other approach is based on sensory properties and volatile components. Wine as a complex matrix contains volatile and non-volatile components that affect the perception of aromas, taste and mouthfeel and, despite the similar chemical characteristics, sensory characteristics can be very different. Studies by Heymann et al. [52], King et al. [53] and Urvieta et al. [54] revealed the fact that differentiation of wines of the same variety, but of different appellations of origin, can be based on sensory properties. Phenolics have also been proposed as chemical markers to establish cultivar authenticity and geographical origin of grapes [54,55]. Slovakia, however, the small country, presents a heterogeneous environment especially in terms of geological and soil conditions. A reflection of this fact can be found in the chemical composition of grapes cultivated in different wine regions. The acidic character of wine was chosen as a group distribution of carboxylic acids (including malic acid, tartaric acid, lactic acid, etc.) in order to classify South Moravian wines [56].

In our study, the possible influence of wine regions on individual chemical parameters was also statistically evaluated.

The PCA outcome is summarized in the ordination diagram (Figure 2). The classification of wines as a group of malic acid, tartaric acid, total acids, glucose, fructose, total polyphenolics and antioxidant activity was chosen as the experimental object. The PCA analysis showed similarities regarding the wine-growing region and grape varieties. The first two axes (shown in the diagram) explain 61% of the total variation of chemical composition. The similarities were observed in wines from the same region, however, not in all three wine varieties this trend was confirmed. Data obtained from wines of Malokarpatská and Južnoslovenská wine regions showed similarities. Clearly distinguished wine-growing regions were Nitrianska and Stredoslovenská. The differences in chemical parameters among these wine regions in relation to the wines indicate the extent to which the subgroups share similar characteristics of wine acidity and sweetness. The most important factor that presented the main descriptor parameter for the distinction among wine-growing regions was the content of organic acids (total acids, tartaric, malic acid and lactic acid). The acid composition and their concentration were also found to correlate with the grape variety, region, processing techniques (especially alcoholic and malolactic fermentation) and aging process [56,57].

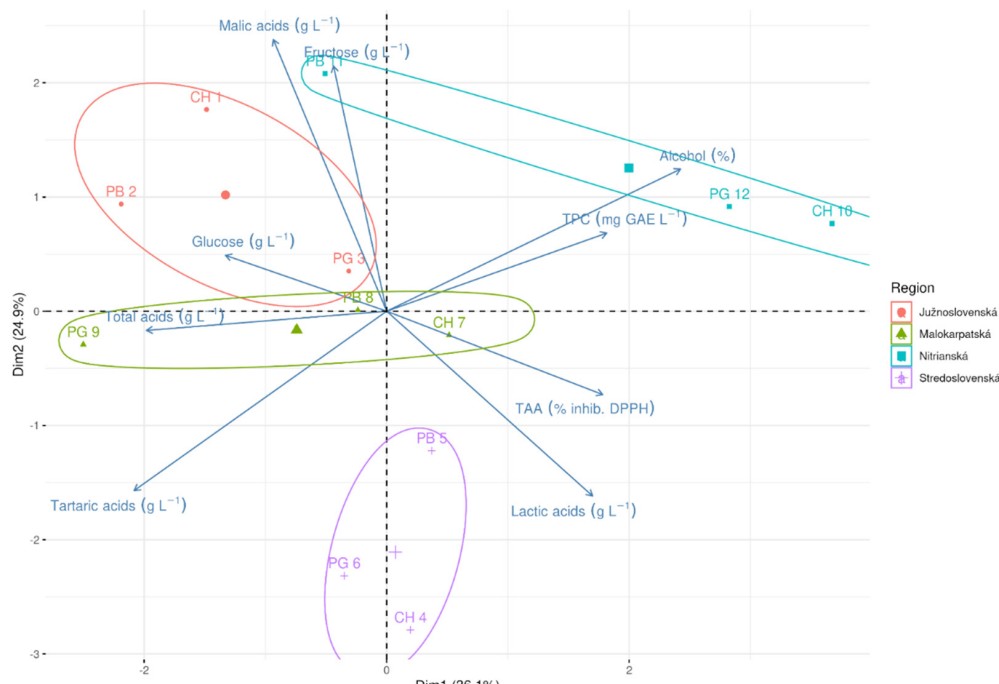

**Figure 2.** PCA bi-plot of wines with chemical parameters grouped by wine regions.

Statistical tests confirmed significant differences in selected chemical parameters in relation to the wine-growing regions (Figure 3). Differences among regions were confirmed in total acids ($F_{3,8}$ = 6.7, $p$ = 0.0142), caused mainly by differences between the Nitrianska and Južnoslovenská wine regions ($t_8$ = 4.37, $p$ = 0.01). Content of tartaric acid was also statistically different ($F_{3,8}$ = 9.8, $p$ = 0.0046) due to the low content in the Nitrianska wine region compared to the values in Južnoslovenská ($t_8$ = 4.2, $p$ = 0.013) and Stredoslovenská wine region ($t_8$ = −5.1 $p$ = 0.004). In the case of malic acid ($F_{3,8}$ = 21.7, $p$ = 0.0003), significant differences were discovered between Južnoslovenská and every other region ($t_8$ = 5.1; 3.3; 7.9, $p$ < 0.05, for Malokarpatská, Nitrianska and Stredoslovenská, respectively) as well as between Nitrianska and Stredoslovenská wine regions ($t_8$ = 4.6 $p$ = 0.008). Content of alcohol was also statistically different ($F_{3,8}$ = 19.208, $p$ = 0.0005) due to the difference between Južnoslovenská and Nitrianska region ($t_8$ = −5, $p$ = 0.013), Malokarpatská and Nitrianska region ($t_8$ = −6.77, $p$ = 0.0006), and Nitrianska and Stredoslovenská region ($t_8$ = 6.3, $p$ = 0.001). Lastly, significant differences were also confirmed in glucose levels ($F_{3,8}$ = 10.8, $p$ = 0.0035), mainly between the Južnoslovenská and Nitrianska wine regions ($t_8$ = 3.4, $p$ = 0.04), the Južnoslovenská and Stredoslovenská wine regions ($t_8$ = 3.5, $p$ = 0.03), the Malokarpatská and Nitrianska wine regions ($t_8$ = 4.4, $p$ = 0.009) and the Malokarpatská and Stredoslovenská wine regions ($t_8$ = 4.5, $p$ = 0.008).

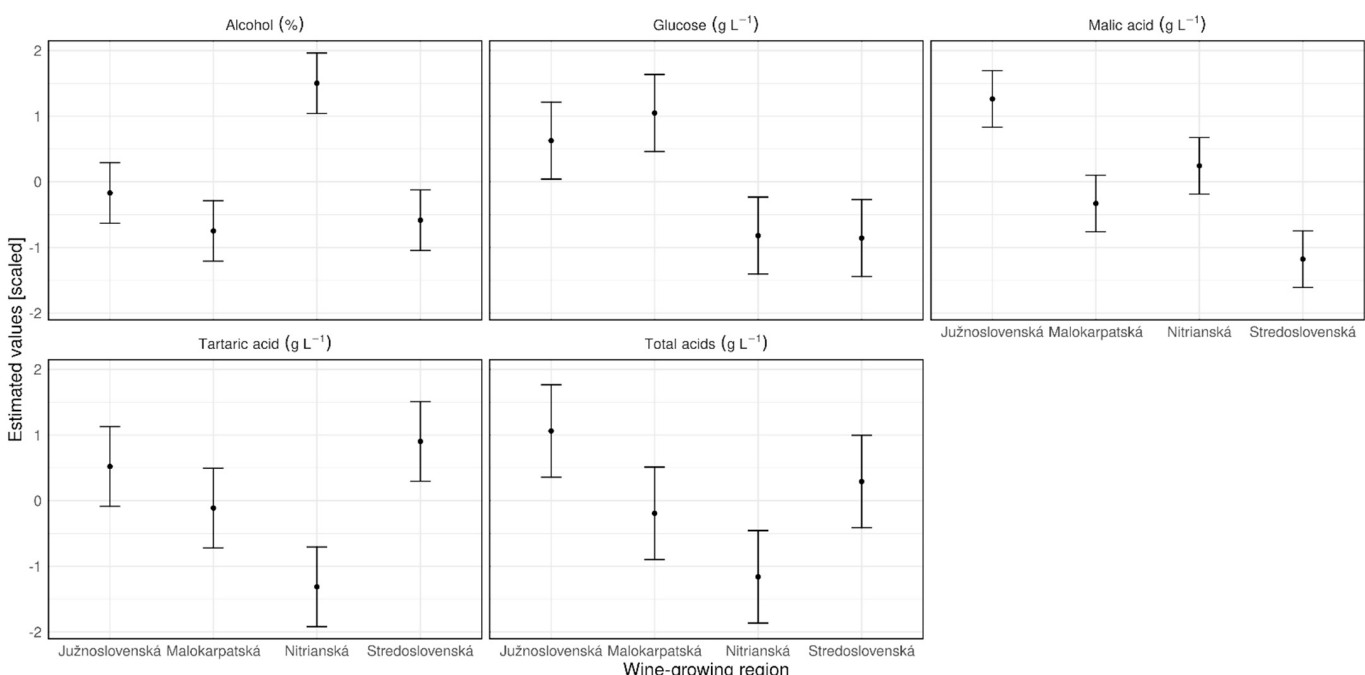

**Figure 3.** Estimated marginal means of scaled data with 95% CI.

## 4. Conclusions

Over the last decades, the study of the plant metabolites of both grapes and wines has been the subject of many investigations. The importance of the approach used is emphasized since the regional or varietal origin of grapes and wines is needed to be identified. An approach that would allow an easy and reliable chemical identification of wines is still the object of interest. Our study was oriented on chemical composition of wines based on determination of alcohol content, main carbohydrates, organic acids and total phenolics and antioxidant activity. The impact of grape varieties and wine-growing regional conditions on selected chemical characteristics of white wine was studied. Correlations between the parameters was examined, we observed high positive correlation between total acids and tartaric acid content, alcohol and total polyphenols, between TPC and TAA was found a weak positive correlation. Negative correlation was high between tartaric acid and alcohol content and between malic and lactic acid. The results showed differences among the regions based on the above-mentioned chemical components. From the parameters that were used in our evaluation, organic acids, content of alcohol and glucose were found to be the most relevant in the classification of the wines. Statistical evaluation of the relationship among chemical parameters, wine variety and wine-growing region showed significant differences, especially in wines from two regions (the Stredoslovenská and Nitrianska wine regions). The white wines are considered to have lower antioxidant activity and content of polyphenols, but in the wines from four Slovak wine regions a high antioxidant activity was confirmed along with the high content of polyphenols. The content of the plant metabolites depends on several factors—grapes variety, region, weather, processing—thus evaluation based on chemical parameters is a complex procedure.

**Author Contributions:** Conceptualization, S.J. and V.I.-P.; data curation, S.J., Z.A., A.M., J.K. and M.Š.; investigation, S.J., I.J. and M.F.; methodology, S.J., A.M. and M.Š.; writing—original draft, S.J. and V.I.-P.; visualization, I.J.; writing—review and editing, M.F. and V.I.-P. All authors have read and agreed to the published version of the manuscript.

**Funding:** This work was funded by Slovak Research and Development Agency (APVV-19-0180), and by the Ministry of Education, Science, Research and Sport of the Slovak Republic grants (KEGA 017/SPU-4/2019 and VEGA 1/0239/21).

**Institutional Review Board Statement:** Not applicable.

**Informed Consent Statement:** Not applicable.

**Data Availability Statement:** Authors are able to make data available on request through the authors themselves. For data requests, the contact details of the corresponding author is present in the affiliation part of the manuscript.

**Acknowledgments:** We would like to thank to the Central European Exchange Program for University Studies, the CEEPUS network CIII-SK-1516-02-2122 entitled BioScience, Food and Health. The AgroBioTech Research Centre in Slovak Agricultural University in Nitra is greatly acknowledged for analysing wine samples by FTIR technique.

**Conflicts of Interest:** The authors declare that there are no conflicts of interest regarding the publication of this paper.

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
