# Peer review of "Chemical Composition of White Wines Produced from Different Grape Varieties and Wine Regions in Slovakia"

_applsci, doi:10.3390/app112211059_

Round 1

Reviewer 1 Report

The paper titled "Chemical composition of white wines produced from different grape varieties and wine regions" by Silvia Jakabová et al. presents and discusses chemical characterization of some Slovak white wines from different regions and grape varieties, using FT-IR and UV-vis based analytical approaches, and performing statistical analysis of experimental data.

The paper is generally well written and experimental methods appear to be sound. However, I have several suggestions and comments that should be carefully addressed before this paper can be considered acceptable for publication.

The first criticism is sample numerosity. The authors state they have selected 12 white wines. It is not specified if these samples correspond to 12 bottles of wines. If so, sample numerosity is very low and too scarce in order to address multivariate analysis of experimental data. Moreover, replicates are not actually real replicates, but pseudoreplicates of different analytical samples from the same bottle.

The second comment is on the chemical compounds quantified. I see lactic acid was not quantified. Is there an explanation for this? lactic acid is a key component of wine. Authors discuss malolactic fermentation, but experimental data lack of a critical parameter, lactic acid. In this regard, malolactic fermentation is generally avoided in white wines, the reason why it is so resides in the fact that the decrease in acidity takes away the taste of "freshness" and floral and fruity bouquet that mostly appreciated in white wines. Can the authors comment on that?   

Thirdly, Table 2 reports concentration values derived from FTIR analyses. However, some of the data reported are out of the range of calibration curves (see lines 148-153, page 4). How did the authors evaluate out-of-range data?

TPC and TAA are expected to be highly related, as the same authors state. However, only a very weak correlation was found. Is there a convincing explanation for that?

Finally, dendrogram (Fig 1) and PCA (Fig 2) are not convincing at all. Dendrogram basically show that data are not easily clusterized. PCA suffers from the scarce sample numerosity.

There are also some minor points.

- English could be improved (uniform verb tenses, punctuation, unintelligible incomplete phrases)

- line 138 page 4 please convert rpm to g

- Line 145 page 4 "the analyzer was calibrated for wine" please specify what parameters are changed by this setting?

- Line 282 page 7 "TPC levels". Since TPC stands for Total Polyphenolic Content, the word "level" should be removed.

Author Response

Author's Reply to the Review Report (Reviewer 1)

We would like to thank Reviewer 1 for his/her comments and suggestions for manuscript improvement. We tried to correct the manuscript according to them.

The paper titled "Chemical composition of white wines produced from different grape varieties and wine regions" by Silvia Jakabová et al. presents and discusses chemical characterization of some Slovak white wines from different regions and grape varieties, using FT-IR and UV-vis based analytical approaches, and performing statistical analysis of experimental data.

Point 1

The paper is generally well written and experimental methods appear to be sound. However, I have several suggestions and comments that should be carefully addressed before this paper can be considered acceptable for publication.

Answer 1

We would like to thank Reviewer 1 for his/her comments and suggestions for manuscript improvement. We tried to correct the manuscript according to them. In the article, we made changes to improve it (marked in tracking changes). If any other improvement is required, we would appreciate to describe them.    

Point 2

The first criticism is sample numerosity. The authors state they have selected 12 white wines. It is not specified if these samples correspond to 12 bottles of wines. If so, sample numerosity is very low and too scarce in order to address multivariate analysis of experimental data. Moreover, replicates are not actually real replicates, but pseudoreplicates of different analytical samples from the same bottle.

Answer 2

12 different wines were obtained from the wine producers, each kind of wine in 3 bottles of the same batch. Wine samples were obtained in June 2020 and analyses were performed on July 1, 2020. Samples were analyzed in triplicates, however, in this number of analyses, statistical evaluation has to be performed with the use of repeated measurements.

Point 3

The second comment is on the chemical compounds quantified. I see lactic acid was not quantified. Is there an explanation for this? lactic acid is a key component of wine. Authors discuss malolactic fermentation, but experimental data lack of a critical parameter, lactic acid. In this regard, malolactic fermentation is generally avoided in white wines, the reason why it is so resides in the fact that the decrease in acidity takes away the taste of "freshness" and floral and fruity bouquet that mostly appreciated in white wines. Can the authors comment on that?   

Answer 3

There were more parameters analyzed by FTIR, including lactic acid and alcohol content. We added these data in table 2. Malolactic fermentation is present in the winemaking process of red wines, and in the production of white wines, this kind of fermentation is not desirable. We agree that in white wines it is avoided due to the negative impact on the sensorial parameters of wines and the wines lose the freshness and floral and fruity bouquet. We agree with this comment, we corrected part of the introduction and discussion and omitted information on malolactic fermentation due to it does not fit so well with the topic of white wines. 

Point 4

Thirdly, Table 2 reports concentration values derived from FTIR analyses. However, some of the data reported are out of the range of calibration curves (see lines 148-153, page 4). How did the authors evaluate out-of-range data?

Answer 4

Thank you for the comment and observation, we corrected the data in the table. The mention data were below the range of the calibration curve. We apologize for the mistake. In the case of total polyphenols, a low injection volume of sample is used for the analysis followed by calculation of TPC, so the content of TPC measured in volumetric flasks was always in the calibration range.

Point 5

TPC and TAA are expected to be highly related, as the same authors state. However, only a very weak correlation was found. Is there a convincing explanation for that?

Answer 5

We agree with this comment. According to the literature sources, we expected a higher correlation of these two parameters, however, the results did not prove this. Analyzing the data, in some samples (CH1, PG6, PB8) we observed a different trend, connected with high TPC values and low TAA, or low TPC and very high TAA. But this trend was confirmed in all three bathes of the wine samples, so we had to include this data in the evaluation.

According to the literature, polyphenolic compounds have shown a different behavior towards DPPH free radical, both in terms of capacity and rate of scavenging (e.g. Villaño, et al., 2007; de Quirós et al., 2009). The antioxidant activity of phenolic compounds is related to their chemical structure; it has been reported that compounds with a high number of hydroxyl groups present higher activity. The contribution of each polyphenol to the antioxidant activity of wines is different, so the activity of wines depends on their phenolic profile. A good correlation was observed between quercetin contents and antioxidant activity. Rutin and Procyanidin B1 showed a reasonable correlation (approx. 0.55); whereas no correlation was found between caftaric acid, the predominant major phenolic compound in wines (de Quirós et al., 2009). Information that white wines can contain a strong antioxidant ascorbic acid that was artificially added to avoid browning was also reported (De Beer et al., 2003).

In our study, no polyphenolic profile nor the presence of artificially added antioxidants in samples was determined.

References cited in the answer:

  • Villaño, D., Fernández-Pachón, M.S., Moyá, M.L., Troncoso, A.M. and García-Parrilla, M.C., 2007. Radical scavenging ability of polyphenolic compounds towards DPPH free radical. Talanta, 71(1), pp.230-235.
  • de Quirós, A.R.B., Lage-Yusty, M.A. and López-Hernández, J., 2009. HPLC-analysis of polyphenolic compounds in Spanish white wines and determination of their antioxidant activity by radical scavenging assay. Food Research International, 42(8), pp.1018-1022.
  • De Beer, D., Joubert, E., Gelderblom, W. C. A., & Manley, M. (2003). Antioxidant Activity of South African Red and White Cultivar Wines: Free Radical Scavenging. Journal of Agricultural and Food Chemistry, 51(4), 902–909. doi:10.1021/jf026011o

Point 6

Finally, dendrogram (Fig 1) and PCA (Fig 2) are not convincing at all. Dendrogram basically show that data are not easily clusterized. PCA suffers from the scarce sample numerosity.

Answer 6

According to the comments of both reviewers, two new parameters were included in the statistical evaluation – the content of alcohol and the content of lactic acid. New results of PCA and dendrogram were recorded in the corrected manuscript. These two parameters improved the observation as is shown in the corrected data in the manuscript. The similarities were observed in wines from the same region, however, not in all three wine varieties, this trend was confirmed. We observed differences in the wines from Stredoslovenská and Nitrianska w.r., but data obtained from wines from in Malokarpatská w.r. and Južnoslovenská w.r. showed similarities, however, both mentioned groups are now more separated as is shown in the PCA and dendrogram.

Point 7

There are also some minor points.

- English could be improved (uniform verb tenses, punctuation, unintelligible incomplete phrases)

Answer 7

English was checked by Proofreading Services.

Point 8

- line 138 page 4 please convert rpm to g

Answer 8

We corrected it, see changes in the text of the manuscript.

Point 9

- Line 145 page 4 "the analyzer was calibrated for wine" please specify what parameters are changed by this setting?

Answer 9

We confirm that this sentence probably is not clear to the reader. We rewrote the sentences as the setting consists only of the choice of type of measurements in the analyzer. Analyzer itself was calibrated by OK Service BioPro, Ltd. (the company provides the apparatus and also technical support), based on the calibration data from the accredited (DAkkS) Institute Heidger (Osann Monzel, Germany). The calibration data consists of over 2000 red and white wines from more than 60 different grape varieties. Software of FT-IR analyzer allows to choose two options of measurements -  wine or must, background signal level is set by measurements of distilled water and accuracy of measurement is checked by measurement of wine with certified values of individual parameters, confirmed by standard reference methods, described in L.162 in the corrected version of the manuscript. The measurement and analysis are then performed fully automatically.

Point 10

- Line 282 page 7 "TPC levels". Since TPC stands for Total Polyphenolic Content, the word "level" should be removed.

Answer 10

We agree, it is deleted in the text.

Reviewer 2 Report

General comments:

This manuscript is clear, well structured, and written. It is relevant for winery, although its impact is locally (Slovakia).

Specific comments will be attached.

Author Response

Author's Reply to the Review Report (Reviewer 2)

We would like to thank Reviewer 2 for his/her comments and suggestions for manuscript improvement. We tried to correct the manuscript according to them.

General comments:

This manuscript is clear, well structured, and written. It is relevant for winery, although its impact is locally (Slovakia).

Specific comments will be attached.

Title: Chemical composition of white wines produced from different grape varieties and wine regions. General comments: This manuscript is clear, well structured, and written. It is relevant for winery, although its impact is locally (Slowakia).

Specific comments:

Point 1

TITLE:

L.1-2: The title must indicate that the wine regions are in Slovakia.

Answer 1

We agree and now the title is corrected.

Point 2

INTRODUCTION:

L.39-40: The part “as well as ….. of chemicals [1], [4].” Is not clear and should be rewritten.

Answer 2

Part is rewritten, see changes in the text.

Point 3

L.46-49: Comment: Maceration of white wines can be done to increase aroma extraction into the must.

Answer 3

We agree, that maceration can increase the aroma components in white wines, however, also the mentioned problem, connected with higher levels of hydroxycinnamates and further formation of undesirable off-odors can occur.

Point 4

L.64-65: Skip “(glucose / fructose = 1:1)”.

Answer 4

We agree with the suggestion, the change is done in the text.

Point 5

L.72-73: Are there any examples or data of wine adulteration by adding glucose or fructose?

Answer 5

The literature source provides information that the NMR method is useful in the detection of the botanical origin of carbohydrates in wine. NMR is used in the control of wine origin, its adulteration, and unauthorized enrichment with sucrose.

Some examples of data about detection of adulteration of wines by adding natural sweeterners:

  • Savin, C., Măntăluţă, A., Vasile, A. and Paşa, R., 2011. Natural or synthetic sweeteners, source of wine adulteration i. Studies on medium-sweet wine adulteration by adding natural sugars to marketable wines. Cercetari Agronomice in Moldova Vol. XLIV, (1).
  • Moro, E., Majocchi, R., Ballabio, C., Molfino, S. and Restani, P., 2007. A rapid and simple method for simultaneous determination of glycerol, fructose, and glucose in wine. American journal of enology and viticulture, 58(2), pp.279-282.

Point 6

L.85-86: Skip “followed with increased concentration of lactic acid”.

Answer 6

We agree with the suggestion, the change is done in the text.

Point 7

L.90-91: Is the formation of tartaric crystals advantageous? Please, comment.

Answer 7

Tartaric acid is the main representative of acids in wine and contributes significantly to the acidity of the wine. Its precipitation in lower temperature into tartaric crystals contributes to decreasing of wine acidity. The presence of tartaric crystals does not influence wine quality but its appearance. It is desirable to get rid of the precipitate by pouring the wine into a carafe or pouring it carefully into the glass so that the precipitate does not swirl in wine.

Point 8

L.98: Do you mean grape composition or grape cultivar composition (blends)?

Grape cultivar composition ?

Answer 8

We mean grape composition. Since the aim of the work was determination of wine composition, but not grape composition, in the next studies we will work on considering grape composition, including various parameters, such as polyphenols, organic acids, carbohydrates.

Point 9

L.102: Use “red varieties” instead of “blue varieties”.

Answer 9

We agree with the suggestion, the change is done in the text.

Point 10

MATERIALS AND METHODS

L.130: The date of sample recollection should be given. This gives an idea about wine age.

Answer 10

We agree with the suggestion, the change is done in the text, L. 127 and 131.

Point 11

L.131: Table 1 is fine and indicates Experimental Design.

L.133-134: Geographic coordinates of each winery should be given.

Answer 11

We added the map with location of wine producers - Fig. 1.

Point 12

RESULTS AND DISCUSSION

L.227: It is a little bit strange that no information about alcohol degree of white wines was given.

Answer 12

We agree with the suggestion, the change is done in table 2.

Point 13

L.247: Use “decreased” instead of “decreases”.

Answer 13

We agree with the suggestion, the change is done in the text.

Point 14

L.314: “Slovakia, however the small country,”. Please check spelling.

Answer 14

We agree with the suggestion, the change is done in the text.

Point 15

CONCLUSIONS

L.366: The conclusions should be stated more specifically. For example: “As there is an increasing interest in natural compounds that have antioxidant, antimicrobial and anti-inflammatory properties, plant metabolites such as organic acids and polyphenols are very preferable”. This is more a discussion than a conclusion. The conclusion must give an answer to the objectives (1) and (2) (L.108 and L.114) stated in the Introduction.

Answer 15

In the article, we made changes to improve it. The conclusion was prepared as was suggested. Changes are marked in tracking changes and with red color. If any other improvement is required, we would appreciate to describe them.    

Round 2

Reviewer 1 Report

The paper has been revised according to many of my previous suggestions, and I believe it has been substantially improved. However, there are still some important points that still need to be addressed in my opinion.

  1. Low correlation value between TPC and TAA has not been sufficiently explained. Authors should more clearly discuss this point in the text.
  2. PCA and dendrogram still appear unuseful in my opinion. Indeed, I recall, I said that each PC in PCA explain less than 50% of variance, and a total variance of 60% is explained by PC1 and PC2, which is very low. Therefore, clusterization shown in Figure 3 is quite misleading. The same can be said for the dendrogram in Figure 2: it shows that samples are not easily clusterized.

Author Response

Author's Reply to the Review Report (Reviewer 1)

We would like to thank Reviewer 1 for his/her comments and suggestions for manuscript improvement. We tried to correct the manuscript according to them. In the article, we made changes to improve it (marked in tracking changes and blue colour).

The paper has been revised according to many of my previous suggestions, and I believe it has been substantially improved. However, there are still some important points that still need to be addressed in my opinion.

Point 1

Low correlation value between TPC and TAA has not been sufficiently explained. Authors should more clearly discuss this point in the text.

Answer 1

We would like to thank Reviewer 1 for his/her comments and suggestions for manuscript improvement. In the text of the manuscript, we added new parts (marked with the tracking changes and blue color) with the explanation of the low correlation between TPC and TAA observed in our results.  In the list of references, three new references were added in order to support the explanation.

Point 2

PCA and dendrogram still appear unuseful in my opinion. Indeed, I recall, I said that each PC in PCA explain less than 50% of variance, and a total variance of 60% is explained by PC1 and PC2, which is very low. Therefore, clusterization shown in Figure 3 is quite misleading. The same can be said for the dendrogram in Figure 2: it shows that samples are not easily clusterized.

Answer 2

We agree that Figure 2 did not provide a clear distinction between wine regions, although some similarities were observed. We decided to omit the dendrogram from the manuscript. Statistical tests confirmed significant differences in selected chemical parameters in relation to wine regions. PCA similarities are shown, but with a total variance of 61%. All available important information from the statistical evaluation is given in the text of the manuscript and in this respect, the information should not be misleading.